# *Phytophthora sojae* Effector PsAvh113 Targets Transcription Factors in *Nicotiana benthamiana*

**DOI:** 10.3390/jof10050318

**Published:** 2024-04-27

**Authors:** Shuai Wu, Jinxia Shi, Qi Zheng, Yuqin Ma, Wenjun Zhou, Chengjie Mao, Chengjie Chen, Zhengwu Fang, Rui Xia, Yongli Qiao

**Affiliations:** 1MARA Key Laboratory of Sustainable Crop Production in the Middle Reaches of the Yangtze River, College of Agriculture, Yangtze University, Jingzhou 434025, China; 2Shanghai Key Laboratory of Plant Molecular Sciences, College of Life Sciences, Shanghai Normal University, Shanghai 200234, China; 3State Key Laboratory for Conservation and Utilization of Subtropical Agro-Bioresources, College of Horticulture, South China Agricultural University, Guangzhou 510640, China

**Keywords:** effector, PsAvh113, *P. sojae*, RNA-seq, VIVE

## Abstract

*Phytophthora sojae* is a type of pathogenic oomycete that causes *Phytophthora* root stem rot (PRSR), which can seriously affect the soybean yield and quality. To subvert immunity, *P. sojae* secretes a large quantity of effectors. However, the molecular mechanisms regulated by most *P. sojae* effectors, and their host targets remain unexplored. Previous studies have shown that the expression of PsAvh113, an effector secreted by *Phytophthora sojae*, enhances viral RNA accumulations and symptoms in *Nicotiana benthamiana* via VIVE assay. In this study, we analyzed RNA-sequencing data based on disease symptoms in *N. benthamiana* leaves that were either mocked or infiltrated with PVX carrying the empty vector (EV) and PsAvh113. We identified 1769 differentially expressed genes (DEGs) dependent on PsAvh113. Using stricter criteria screening and Gene Ontology (GO) and Kyoto Encyclopaedia of Genes and Genomes (KEGG) analysis of DEGs, we found that 38 genes were closely enriched in response to PsAvh113 expression. We selected three genes of *N. benthamiana* (*NbNAC86*, *NbMyb4*, and *NbERF114*) and found their transcriptional levels significantly upregulated in *N. benthamiana* infected with PVX carrying PsAvh113. Furthermore, individual silencing of these three genes promoted *P. capsici* infection, while their overexpression increased resistance to *P. capsici* in *N. benthamiana*. Our results show that PsAvh113 interacts with transcription factors NbMyb4 and NbERF114 in vivo. Collectively, these data may help us understand the pathogenic mechanism of effectors and manage PRSR in soybeans.

## 1. Introduction

Soybean (*G. max*) is one of the essential food crops in China and occupies an important position in the agricultural industry structure. During the growth of soybeans, various biotic and abiotic stresses in the environment can significantly impact both the yield and quality of soybeans. Oomycetes, a lineage of eukaryotic microorganisms within the kingdom Stramenopila, encompass pathogens affecting plants and animals [1,2,3]. Numerous devastating crop pathogens belong to the genus *Phytophthora* sp. [3,4,5]. Among them, *P. sojae*-induced root rot has significantly impacted soybean production, and in severe cases, it can even result in crop failure. The global annual loss from soybeans caused by *Phytophthora* sp. pathogens has been estimated at over USD 2 billion [4]. The infection of *P. sojae*, *Botrytis cinerea*, *Fusarium oxysporum*, and *Rhizoctonia solani* causes soybean root rot, with *P. sojae* being one of the most extensively reported and studied pathogens worldwide. The plant pathogen *Phytophthora* sp. is the most devastating threat to agricultural production and the natural environment [6]. The host range of *P. sojae* is highly restricted; however, it can infect soybeans throughout their entire growth period. It exhibits high incidence rates, short incubation periods, rapid transmission speeds, and the ability to cause multiple infections, demonstrating its strong pathogenicity.

There is a perpetual arms race between plants and microbial pathogens, and the classical zig-zag model fully illustrates the host–pathogen relationship [7]. In the cytoplasmic space, a fierce battle of attack, counter-attack, and counter-counter-attack ensues, often determining the outcome of pathogen–plant interactions [7,8]. The pathogens initiate infection and colonization within the plant by inducing mutations in the primary defense mechanism of the plant cell wall. However, plants have evolved diverse strategies to cope with pathogen infection. Plant pattern recognition receptors (PPRs) can recognize PAMPs on pathogens to resist further infection, a process called PTI immune response. The plant PTI response typically accompanies programmed cell death, an outbreak of reactive oxygen species (ROS), and other associated phenomena. The PTI reaction does not always impede the infection process of pathogens. The latter may secrete effectors that interfere with the PTI reaction to invade host cells further and enhance plant susceptibility. At the same time, plants responded by evolving intracellular receptor proteins, NLRs, which can specifically recognize specific effectors, thus initiating a more robust ETI immune response [7,9,10].

The *Phytophthora* RxLR effectors are well-studied in plants, primarily known for its N-terminal region containing the conserved RxLR-dEER domain [11,12]. The infection of plants by pathogens triggers programmed cell death, and *Phytophthora sp* typically secrete a substantial quantity of RxLR effectors to suppress the programmed cell death in plants [13], thereby facilitating pathogen invasion into host plants. The extensive study of RxLR effectors stems from their ability to employ sophisticated pathogenicity mechanisms, targeting plant proteins and disrupting host immunity. Some RxLR effectors also trigger immune responses mediated by plant NLRs [14,15,16]. 563 RxLR effectors have been identified in pathogenic *Phytophthora infestans* [5], with approximately 400 RxLR effectors detected specifically in soybean *P. sojae*. The study revealed that most RxLR effectors exhibited upregulation during the early stages of *P. infestans*, thereby highlighting their pivotal role in the process of *P. infestans*. The study revealed that most RxLR effectors exhibited upregulation during the early stages of *P. infestans*, thereby highlighting their pivotal role in the process of *Phytophthora infestans*. The reported findings suggest that PsAvh113 significantly promotes *P. sojae* in plants while demonstrating its ability to suppress *GmCAT1*-induced cell death through binding with GmDPB. Consequently, this interaction ultimately enhances the susceptibility of plants to *P. sojae* [6].

*NbNAC86* is involved in transcriptional regulation and DNA templated. In *Arabidopsis*, its homologs ANAC071 and ANAC096, the NAC-multiple mutants exhibited a significant reduction in wound-induced cambium formation in incised stems. They suppressed the conversion from mesophyll cells to cambial cells. ANAC071 and ANAC096 redundantly participate in the process of “cambialization”, which involves transforming differentiated cells into cambial cells. These proliferating cambium-like cells contribute to tissue regeneration during wound healing [17]. As a transcription factor, AtMyb13, the NbMyb4 homolog in *Arabidopsis*, regulates gene expression by binding to promoters and is involved in auxin response and flavonoid biosynthesis. The synergistic effect of UVR8, a UV-B photoreceptor, with various transcription factors in the nucleus coordinates the expression of specific downstream genomes. It ultimately mediates plant response to UV-B light [18]. NbERF114 is involved in the ethylene signaling pathway. Its homologous gene AtERF115 in *Arabidopsis* represses ARI by activating cytokinin (CK) signaling. Additionally, AtERF115 is transcriptionally activated by jasmonate (JA) [19].

We previously established a virus-induced virulence effector assay (VIVE) [20]. *P. soaje* virulence effector PsAvh113 binds to transcription factor GmDPB and inhibits *GmCAT1*-induced cell death, subsequently increasing plant susceptibility to *P. sojae* in soybean [6]. In this study, we analyzed RNA-sequencing (RNA-seq) data based on disease symptoms in *N. benthamiana* leaves mock and infiltrated with PVX carrying the empty vector (EV) and *PsAvh113*, identified many DEGs in response to PsAvh113. Furthermore, we found 38 genes enriched in response to *PsAvh113* expression. We selected three genes (*NbNAC86*, *NbMyb4,* and *NbERF114*) and found that these genes positively regulated the resistance to *P. capsici* in *N. benthamiana.* Our results show that PsAvh113 is directly associated with transcription factors NbMyb4 and NbERF114 in vivo.

## 2. Materials and Methods

### 2.1. Plants and Microbial Strains

*N. benthamiana* plants were cultivated in a growth chamber under LED lamps, with a 16-h light/8-h dark photoperiod, and maintained at 24–25 °C. The light intensity was set at approximately 120–150 μmol m^−2^s^−1^ [21]. The *P. capsici* (PC35) strain was maintained in 25 °C incubators on V8 juice medium plates under dark conditions for 3 days before inoculation. The yeast strain AH109 was incubated at 30 °C for 3 days for Yeast 2-Hybrid (Y2H) assays. *Agrobacterium* (*Agrobacterium tumefaciens*) strain GV3101 was incubated on LB medium supplemented with 50 μg/mL rifampicin and 50 μg/mL gentamycin at 30 °C overnight. The *Escherichia coli* strain DH5α used for vector construction was cultured on LB medium at a temperature of 37 °C.

### 2.2. Virus-Induced Virulence Effector Assay (VIVE)

The PCR products of PsAvh113 (excluding signal peptides) were inserted into the pGR106 vector, which contains the entire PVX genome. Two-week-old N. benthamiana plants were then infiltrated with Agrobacterium tumefaciens carrying the plasmids mock, pGR106-EV, and pGR106-PsAvh113 mutants at an OD of 0.6. Following a 2–3-week period, the infiltrated plants were visually documented, and total RNA was isolated from these plants using Trizol™ Reagent (Thermo Fisher Scientific, Waltham, MA, USA).

### 2.3. Virus-Mediated Gene Silencing (VIGS) Assay

The DNA fragment was designed by sequencing analysis and then cloned into a TRV2 vector. The resulting constructs were transformed into *Agrobacterium* GV3101 and transiently co-expressed with TRV1 in 10-day-old l *N. benthamiana* leaves. After 2–3 weeks, fresh leaves were collected for RNA extraction, followed by verification of gene silencing efficiency using qRT-PCR. Subsequently, leaves exhibiting significant silencing effects were selected for *P. capsici* infection.

### 2.4. Phytophthora Capsici Infection and Biomass Determination Assays

Plasmids of the gene were transiently expressed in *N. benthamiana* leaves via *A. Agrobacterium*-mediated infiltration, respectively. After 24 h, the leaves were detached and inoculated with about 2000 zoospore suspensions of *P. capsici* isolate PC35. Inoculated leaves were incubated in a growth chamber at 24 °C for 2–3 days before disease progression analysis. The *P. capsici* lesions were measured and photographed under a UV lamp.

Leaf discs of *N. benthamiana* were sampled from the infected site 40–60 h after *P. capsici* inoculation. Pure genomic DNA was extracted using a genomic DNA extraction kit (Tiangen Biotech, Beijing Co., Ltd., Beijing, China). *P. capsici* biomass in inoculated leaves was determined by qPCR using primers specific for *N. benthamiana* and *P. capsici* actin genes (Appendix A).

### 2.5. DNA and RNA Isolation, qRT-PCR

The leaf discs were collected from *N. benthamiana* leaves infected with *P. capsici* at 2 dpi. An equal amount of samples was utilized for genomic DNA extraction using a genomic DNA isolation kit. Total RNA was extracted using TRIzol™ Reagent following the manufacturer’s protocol. A 1 μg amount of total RNA was reverse transcribed with oligo (dT18) primers in a 20 μL reaction volume using the HiScript II 1st Strand cDNA Synthesis Kit (+gDNA wiper) (Vazyme Biotech Nanjing, Nanjing, China). PCR amplification was performed with gene-specific primers (Appendix A). The qRT-PCR was used to assess gene transcript abundance and conducted on a Lightcycler 480II^®^ instrument (Roche, Basel, Switzerland) using the ChamQ Universal SYBR qPCR Master Mix Kit (Tiangen Biotech, Beijing Co., Ltd., Beijing, China). 

### 2.6. Digital RNA-Seq and Data Analysis

The RNA samples from mock, EV, and PsAvh113 were subjected to RNA-seq analysis. Mock represents *N. benthamiana* plants, EV and PsAvh113 individually represent *N. benthamiana* plants infiltrated by *A. tumefaciens* containing the plasmid PVX-EV, and PVX-PsAvh113. RNA-Seq libraries were constructed according to the manufacturer’s procedure. Transcriptome sequencing was performed using the Illumina Hiseq X Ten System (Illumina, San Diego, CA, USA) to generate 150 bp paired-end (PE) raw reads, which were then filtered to remove linker reads and low-quality reads to obtain clean reads. Then, STAR was used to align Clean reads to the reference genome Niben1.0.1 (https://solgenomics.net/ftp/genomes/Nicotiana_benthamiana/assemblies/, accessed on 2 February 2024) of *Nicotiana benthamiana* [22]. After that, StringTie (v2.2.1) was used to assemble and quantify the reads on the alignment [23]. Then, the FPKM (Fragments Per Kilobase of transcript per Million fragments mapped) value was used as an indicator to measure gene expression levels and calculate the expression level of each sample. The differential expression analysis between the sample groups was performed using DESeq2 (v1.42.0) [24]. The genes with |log_2_FoldChange| ≥ 1 and padj ≤ 0.05 were selected as differential genes, and the volcano plot of the differential genes was drawn using the ggrepel (v0.9.5) package in R. The differential gene IDs were visualized by using the Venn diagram generated by TBtools-II (v2.069). Then, the 1769 genes common to PsAvh113_vs_EV and PsAvh113_vs_Mock were extracted for GO (Gene Ontology) and KEGG (Kyoto Encyclopedia of Gene and Genomes) functional enrichment analysis. According to the most significantly enriched entries in the biological process category, the enriched genes were extracted, and the highly expressed genes in PsAvh113 were screened by adopting more stringent criteria (log_2_FoldChange > 4 and padj < 1 × 10^−5^). Then, the heat map of the screened gene expression quantity was drawn by using TBtools-II. For the gene’s function, the screened gene protein sequences were compared in the UniProtKB/Swiss-Prot database to check the function of homologous proteins manually.

### 2.7. Transient Gene Expression in N. benthamiana and Western Blotting

For transient expression analysis, the validated plasmids were introduced into *A. tumefaciens* strain GV3101 and then infiltrated in the leaves of 3–4-week-old *N. benthamiana*. Protein expression was detected by western blotting at 48 hpi. The proteins were separated on SDS-PAGE gels through electrophoresis and transferred onto a PVDF membrane using the semidry electroblotting technique. Subsequently, the membrane was incubated with an anti-HA (MBL, catalog No. 330) at room temperature for 60 min. Finally, signals were developed using chemiluminescence substrate (Thermo Fisher Scientific, USA) and imaged using Amersham^TM^ Image 600 system (GE Healthcare, Pittsburgh, PA, USA).

### 2.8. Bimolecular Fluorescence Complementation (BiFC) Assay

The full-length CDS of genes were individually cloned into the pQBV3 vector and recombined into the pEarleyGate201-YN and pEarleyGate202-YC vectors [25]. The resulting constructs were transformed into *Agrobacterium* GV3101 and transiently expressed in *N. benthamiana* leaves following the method described by Qiao et al. [26]. YFP fluorescent signals were monitored using a Zeiss LSM 710© confocal microscope (Leica, Weztlar, Germany) 48 hpi.

### 2.9. Yeast 2-Hybrid (Y2H) Assay

The full-length cDNA sequences of *NbNAC86*, *NbMyb4*, and *NbERF114* were individually cloned into the bait vector pGBKT7 (Clontech, Mountain View, CA, USA), while the full-length CDS of PsAvh113 was inserted into the prey vector pGADT7. AD-LaminC and AD-SV40T were co-transformed with BD-p53 and served as negative and positive controls, respectively. The resulting bait and prey constructs were co-transformed into *Saccharomyces cerevisiae* AH109 strain. The transformed cells were streaked on SD/–Trp/–Leu medium and incubated at 30 °C for 2 days. Subsequently, the cells were transferred onto a stringent medium (SD/–Trp/–Leu/–His/–Ade). Plates were incubated at 30 °C for 4–8 days before evaluation and photography [27].

## 3. Results

### 3.1. Transcriptome Analysis of N. bentamiana Infected with PVX Carrying EV and PsAvh113

Our previous study showed that *PsAvh113* is a virulence effector, and its expression markedly enhanced viral symptoms and viral RNA accumulation in *N*. *benthamiana* leaves by VIVE assay [6]. To study the transcriptional response of *N*. *benthamiana* to *PsAvh113*, we extracted RNAs from *N*. *benthamiana* mock, infected with PVX carrying EV and PsAvh113, respectively. RNA-Seq libraries were constructed and sequenced with the Illumina Hiseq X Ten System. RNA-Seq of the nine samples (three groups, three duplicates) generated 165.34 million raw reads, resulting in 53,286 transcripts and 53,286 genes in the *N*. *benthamiana* genome. The Pearson correlation among the three biological replicates from each intra-group exceeded 0.93, and the inter-group correlation coefficients of the three biological repeats clustered into one class (Figure 1A). These results indicate that the sequencing data meets the sequencing quality control standard and the requirements for subsequent bioinformatic analyses.

To further understand the transcriptional response of *N*. *benthamiana*, differential expression profiling was performed among each combination using DESeq2 software (v1.42.0) (*p* ≤ 0.05, |log2 Fold Change| > 1). We found 3327 significant DEGs between the EV and mock groups among 10,348 genes (Appendix A), including 1937 upregulated and 1390 downregulated genes; 6215 significant DEGs between the PsAvh113 and mock groups among 10,327 genes (Appendix A), including 3738 upregulated and 2477 downregulated genes; 3710 significant DEGs between the PsAvh113 and EV groups among 10,335 genes (Appendix A), including 1891 upregulated and 1819 downregulated genes (Figure 1B). In addition, we identified 1769 DEGs by Venn diagram, which may specifically cause the disease symptoms by PsAvh113 expression by comparing with EV and mock (Figure 1C). Therefore, our study focused on these 1769 DEGs.

### 3.2. GO and KEGG Enrichment Analysis of DEGs

Functional assignments were defined using GO terms, which provide a comprehensive functional classification of genes and their products based on various biological processes (BP), cellular components (CC), and molecular functions (MF). Therefore, we used DAVID to perform gene enrichment and functional annotation (GO) analysis of the 1769 genes (Appendix A). GO functional enrichment analysis revealed several enriched terms: 20 MF terms, mainly involved in catalytic activity, transferring phosphorus-containing groups, and oxidoreductase activity; 20 CC terms, including cell periphery, plasma membrane, obsolete plastid part, and obsolete chloroplast part; and 74 BP terms, including response to stimulus, response to chemical, response to abiotic stimulus, response to oxygen-containing compound, and phosphorus metabolic. Interestingly, we noticed that the most significant enrichment was a response to stimulus in the BP term, with 423 DEGs (Figure 2A).

Pathway-based analysis was performed using the KEGG pathway database (15 January 2024) to explore the biological functions and interactions in the genes. Our results showed that 1769 DEGs were primarily involved in 32 pathways. The top 10 most significantly enriched pathways were selected to draw scatter plots. The metabolic pathways with significant enrichment were mainly metabolism, including carbohydrate metabolism and biosynthesis of other secondary metabolites (Figure 2B).

To further reveal whether PsAvh113 directly causes some DEGs, we filtered the 423 genes enriched in response to stimulus, adopted stricter criteria (log_2_FoldChange > 4 and padj < 1 × 10^−5^) to identify the genes highly expressed in PsAvh113, and finally aimed at 38 genes (Figure 2C) (Appendix A). After comparing these 38 genes and manually checking the possible functions of these genes, we found that the functions of most genes were related to auxin regulation, and then selected three genes that were used to figure out the direct interaction with PsAvh113, such as *NbNAC86* (*NbD004834*) is a protein with NAC domain, which may be involved in regulating secondary metabolism of plants [28]; *NbMyb4* (*NbD028144*) is a Myb4-like protein, which is involved in the regulation of flavonoid and lignin synthesis [29]; *NbERF114* (*NbD040014*) is an ERF114-like protein, which can respond to ethylene signals and then participate in the response of plants to environmental stress [30].

### 3.3. Validation of Three Gene Expression in PsAvh113-Expressed N. benthamiana by Real-Time qRT-PCR

RNA-seq data showed that the transcriptional levels of these three genes were significantly upregulated in (Figure 3A). To verify the robustness and accuracy of RNA-seq results, we infiltrated *N. benthamiana* plants with PVX carrying EV or PsAvh113 and determined the relative expression of three candidate genes. qRT-PCR data exhibited that the relative transcriptional levels of *NbNAC86, NbMyb4*, and *NbERF114* were upregulated when compared with EV, particularly for *NbNAC86* and *NbERF114* (Figure 3B). This indicates that the RNA-seq outcomes were consistent with qRT-PCR data, and thus, the RNA-seq outcomes were reliable.

### 3.4. NbNAC86, NbMyb4, and NbERF114 Positively Regulate the Resistance to P. capsici in N. benthamiana

To elucidate the potential role(s) of *NbNAC86*, *NbMyb4*, *and NbERF114* in plant immunity, we first investigated the *NbERF114* gene in *N. benthamiana* using virus-induced gene silencing (VIGS) to determine their function in plant defense during *Phytophthora* sp. infection. One VIGS construct (*VIGS1*, *NbERF114i*) with a 300bp length fragment was designed to silence the *NbERF114* gene specifically. *VIGS* construct successfully silenced this target as shown by qRT-PCR analysis, without affecting the growth and development of *N. benthamiana* plants (Appendix A). The *NbERF114*-silenced plants were more susceptible to *P. capsici* than the control plants (Figure 4A), and *P. capsici* lesion length and relative biomass were markedly greater on leaves silenced of *NbERF114* than on those transformed with the empty vector (EV) control (Figure 4B,C). Consistent with the above data, individual silenced of *NbNAC86* or *NbMyb4* in *N. benthamiana* leaves enhanced susceptibility to *P. capsici* as indicated by the relative biomass and lesion length of the *P. capsici*, which were significantly greater on leaves transiently silencing of *NbNAC86* or *NbMyb4* than on those transformed with the *GFP*-containing EV control (Figure 4).

To further determine the functions of these three genes in *N. benthamiana*, we individually cloned the *NbNAC86*, *NbMyb4*, or *NbERF114* genes. Then we introduced them into the plant expression vector pEarlyGate101 with YFP-HA tag, which resulted in NbNAC86-YFP-HA, NbMyb4-YFP-HA, and NbERF114-YFP-HA constructs. Subsequently, we transiently expressed these genes in *N. benthamiana* leaves via *Agrobacterium*-mediated infiltration, respectively. Expression of these proteins was confirmed by Western blotting at 48 hpi (Appendix A). Pathogen inoculation showed that transient expression of *NbNAC86*, *NbMyb4*, or *NbERF114* gene in *N. benthamiana* leaves enhanced resistance to *P. capsici* (Figure 5A) as indicated by the relative biomass and lesion length of the *Phytophthora* pathogen, which were significantly smaller on leaves transiently expressing *NbNAC86*, *NbMyb4*, or *NbERF114* gene than on those transformed with the *GFP*-*HA*-containing EV control (Figure 5B,C). Collectively, these data suggest that *NbNAC86*, *NbMyb4*, and *NbERF114* genes act as positive immune regulators against *P. capsici* infection in *N. benthamiana*.

### 3.5. PsAvh113 Interacts with Both NbMyb4 and NbERF114 In Vivo

To obtain insight into the direct association of PsAvh113 with these immune regulators, we first cloned the CDS sequence of NbERF114 into bait vector pGBKT7 (BD) and PsAvh113 into the prey vector pGADT7 (AD). We performed the Y2H assay (Figure 6A). The yeast strain AH109 was transformed with the BD plasmid carrying NbERF114 and the AD plasmid pGADT7 carrying PsAvh113. The results displayed that yeast cells co-transformed NbERF114 with PsAvh113 could grow on a minimal medium, while yeast cells co-transformed NbERF114 with EV were unable to grow on it, indicating that PsAvh113 interacts with NbERF114 in yeast (Figure 6A). Consistent with the above data, yeast cells co-transformed PsAvh113 with NbMyb4 could grow on a minimal medium (Figure 6A).

To further elucidate the interaction between PsAvh113 and, NbMyb4 and NbERF114, we conducted the bimolecular fluorescence complementation (BiFC) assay to confirm their physical interaction in *N. benthamiana*. The *PsAvh113, NbMyb4,* and *NbERF114* genes were fused to the *YFP* gene sequence corresponding to its N-terminal half (YN) and C-terminal half (YC), respectively, and the resulting constructs were coexpressed in *N. benthamiana* leaves. At 48 hpi, YFP signal was detected in the epidermal cells of leaves expressing *NbERF114-YC* together with *PsAvh113-YN, NbMyb4-YC* together with *PsAvh113-YN*, but not in cells expressing *NbMyb4-YC*+*YN*, *NbERF114-YC-YC*+*YN*, or *YC*+*PsAvh113-YN* (Figure 6B). The YFP signal was mainly localized to the nucleus and cytoplasm. Collectively, these results demonstrate the association of PsAvh113 with both NbMyb4 and NbERF114 in vitro and in vivo. In addition, we also determined the subcellular localization of NbMyb4 and NbERF114 in *N. benthamiana* leaves via *Agrobacterium*-mediated transformation. Confocal microscopy analysis of the agroinfiltrated leaves showed that NbMyb4 localized primarily to the nucleus of leaf epidermal cells, while NbERF114 localized to the cytoplasm and nucleus of leaf epidermal cells (Figure 6C).

## 4. Discussion

Plant disease outbreaks pose significant risks to global food security and environmental sustainability worldwide. One of the main factors hindering the resistant breeding of cereal crops is plant pathogens [31,32]. Increasing evidence shows that *P. sojae* secretes an arsenal of host cytoplasmic effectors that enhance pathogen colonization and disease development by modifying host target functions and disrupting the immune signaling network [8,33]. However, the host targets of most virulence effectors in *Phytophthora* pathogens have not been identified. 

In this study, we utilized a newly developed screening assay, VIVE [20], to analyze RNA-sequencing (RNA-seq) data based on disease symptoms caused by PsAvh113 in *N. benthamiana* leaves and identified 38 DEGs (differentially expressed genes) enriched in response to *PsAvh113* expression. We selected three genes (*NbNAC86*, *NbMyb4,* and *NbERF114*) and discovered that all these genes positively regulate the resistance to *P. capsici* in *N. benthamiana*, both NbMyb4 and NbERF114 have a direct association with PsAvh113.

Plant pathogens use a variety of invasion tactics to successfully invade and propagate in host plants, including secreting many effectors as invasion weapons [31]. Effector proteins often have more than one host target and can target multiple steps in a plant’s signaling pathways [34], while many different effectors can bind to one key target [35,36]. Our previous research found that a virulence effector from *P. soaje*, PsAvh113, binds to GmDPB, a transcription factor, and inhibits *GmCAT1*-induced cell death, making soybean plants more susceptible to *P. sojae* [6]. In this study, we identified two new targets of PsAvh113 by RNA-seq analysis. All three potential targets encode transcription factors that positively affect plant resistance to *P. capsici* in *N. benthamiana*. This indicates that PsAvh113 subverts plant immunity by interacting with various transcription factors in the nucleus to orchestrate the expression of specific genes, ultimately leading to pathogen colonization and disease development. However, we still do not know how NbMyb4 and NbERF114 regulate disease symptoms and viral accumulations by PsAvh113, and further research is needed to understand their biological function in *N. benthamiana*.

The VIVE assay has successfully identified the virulence factor [20]. For instance, the secretory protein SDE3 of “*Candidatus* Liberibacter asiaticus” was identified as a potential virulence effector. It directly binds to the cytosolic CsGAPC1/2 proteins, inhibits host autophagy, and suppresses CsATG8-mediated immunity which promotes Huanglongbing disease in citrus [37]. In this study, the VIVE assay and RNA-seq analysis were used to identify potential molecular targets in plants. We identified 38 genes closely enriched in response to *PsAvh113* expression. More than half of them were related to auxin regulation, which has been reported to regulate pathogen infection [38]. Consequently, we selected three genes and verified the roles of two genes (*NbMyb4* and *NbERF114*) in regulating plant immunity and their interactions with PsAvh113. To our knowledge, we believe that this approach could be used to screen potential interacting proteins. Because most genes respond to effector-dependent expression after removing the background data of EV and mock. 

In a recent study by Qian et al., it was reported that *AtMYB13,* a homolog of *NbMyb4* in *Arabidopsis*, regulates the expression of genes involved in auxin response and flavonoid biosynthesis. This regulation is achieved through direct binding with their promoters, ultimately mediating plant response to UV-B light [18]. Another study by Canher et al. found that *AtERF115*, an *Arabidopsis* homology of *NbERF114*, is a wound-inducible regulator of stem cell division. AtERF115 sensitizes cells to auxin by activating ARF5/MONOPTEROS, an auxin-responsive transcription factor involved in the global auxin response, tissue patterning, and organ formation [39]. Our research focuses on understanding the role of NbMyb4 and NbERF114 in manipulating the expression of downstream genes and disease-resistance of *Phytophthora* root and stem resistance in soybeans. Our current data have shown that NbMyb4 and NbERF114 interact with PsAvh113 and positively regulate plant resistance to *P. capsici* in *N. benthamiana*. However, their specific mechanisms in manipulating the expression of downstream genes and disease-resistance of *Phytophthora* root and stem resistance in soybeans are still unclear. Therefore, we will primarily focus on investigating the associations of PsAvh113 with these two genes based on VIVE assay and RNA-seq analysis. At the same time, there needs to be more in-depth investigation into their role in plant disease resistance and molecular mechanisms.

In conclusion, we investigated the DEGs in response to *P. sojae* effector PsAvh113 expression in *N. benthamiana* and identified three significantly upregulated genes (*NbNAC86*, *NbMyb4*, and *NbERF114*). Subsequently, we furthermore verified that PsAvh113 is directly associated with transcription factors NbMyb4 and NbERF114, which positively regulated the resistance to *P. capsici* in *N. benthamiana*.

## Figures and Tables

**Figure 1 jof-10-00318-f001:**
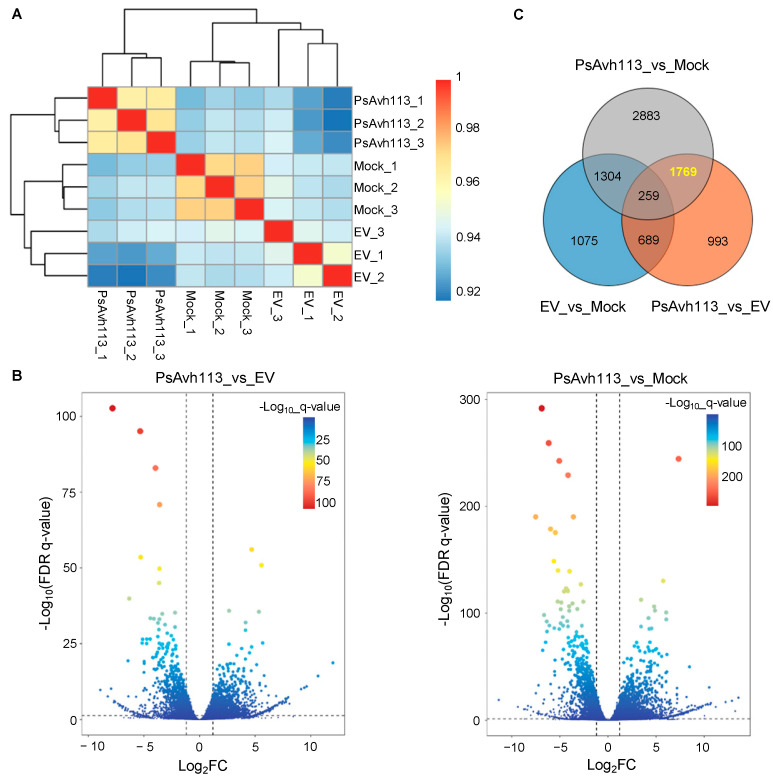
Comparisons of differentially expressed genes (DEGs) in *N. benthamiana* plants. *N. benthamiana* leaves were infiltrated with *Agrobacterium tumefaciens* strain GV3101 carrying EV or PsAvh113, Mock plants were used as the negative control. (**A**) Hierarchical cluster analysis (HCA) of transcriptome data. Color labels from blue to red indicate low to high correlations between samples. (**B**) Volcano plot of significantly differentially expressed genes in Mock, EV, and PsAvh113. Comparisons of EV and PsAvh113 (**left panel**), and Mock and PsAvh113 (**right panel**). FDR, false discovery rate, and FC, fold change. Dashed lines represent significant horizontal lines. (**C**) Venn diagram showing the DEGs identified by comparing EV, Mock, and PsAvh113-Mock. “1769” refers to the identification by Venn diagram of DEGs that may cause disease symptoms through PsAvh113.

**Figure 2 jof-10-00318-f002:**
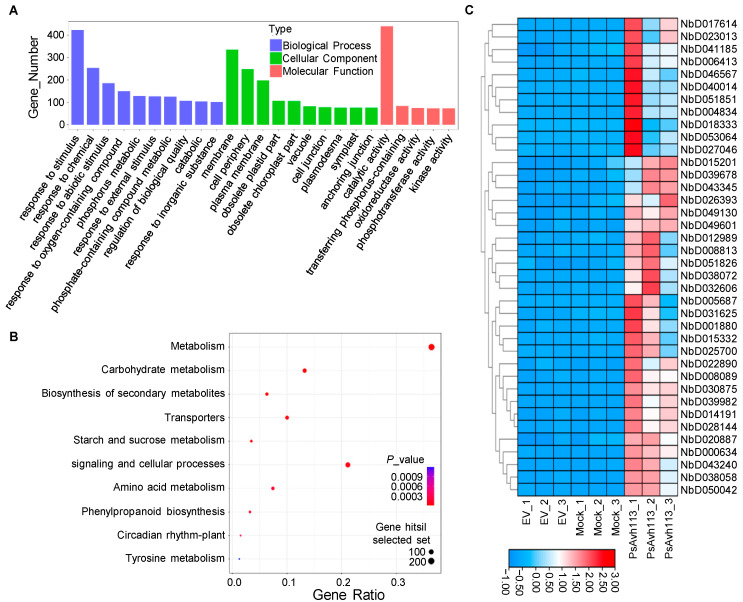
Gene Ontology (GO)-based analysis and Kyoto Encyclopedia of Genes and Genomes (KEGG) enrichment analysis of DEGs. (**A**) Analysis of GO function enrichment in the 1769 DEGs of Avh113. (**B**) Enrichment of differential gene metabolic pathways in the 1769 DEGs of Avh113. Note: The color represents the size of the Q-value, the lower Q-value represents the more significant enrichment, and the dot size represents the number of enrichment differential genes. (**C**) Heat maps of significantly differentially expressed genes (DEGs) between EV, Mock, and PVX-PsAvh113 were generated by RNA-seq analysis. A total of 1769 DEGs were identified using *p*-adjust < 1 × 10^−5^ and log_2_ Fold Change ≥ 4 as screening criteria.

**Figure 3 jof-10-00318-f003:**
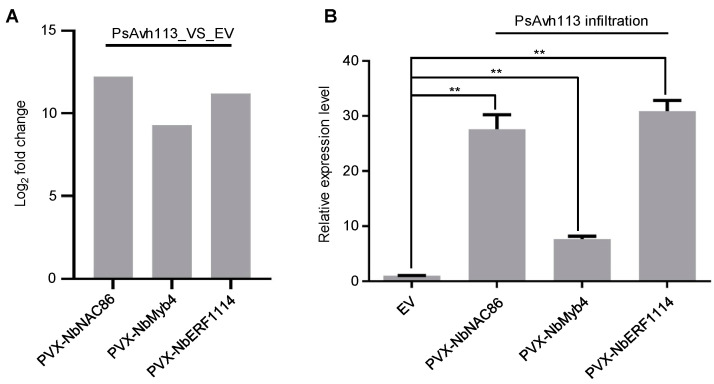
PsAvh113 induces the upregulated expression of *NbNAC86*, *NbMyb4*, and *NbERF114* genes in *N. benthamiana* leaves inoculated with *Agrobacterium tumefaciens* carrying EV or PsAvh113. (**A**) The fold change in 3 DEGs genes in RNA-Seq data. (**B**) The relative expression levels of target genes were determined by qRT-PCR, Data represent mean ± SE of three independent experiments. The asterisk indicates that there is a statistically significant difference in the *t*-test (** *p* < 0.01).

**Figure 4 jof-10-00318-f004:**
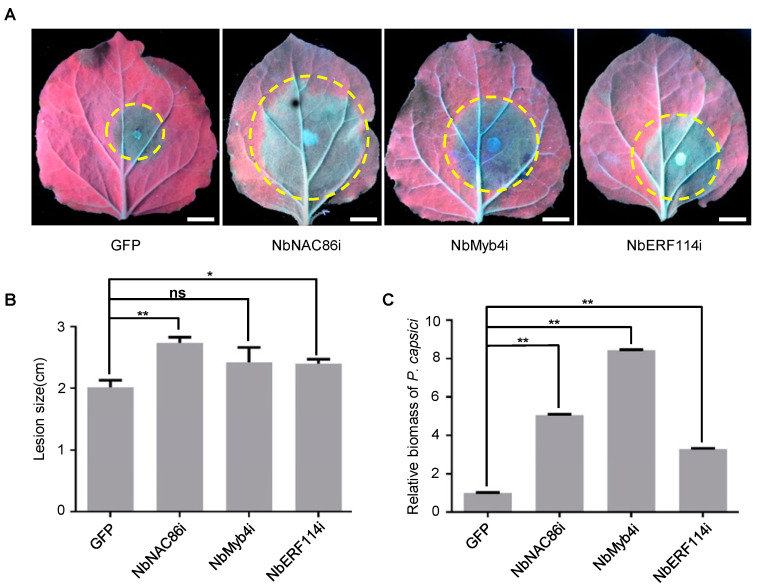
Silencing of *NbNAC86*, *NbMyb4*, or *NbERF114* promotes *P.capsici* infection in *N. benthamiana* leaves, respectively. (**A**) Symptoms of *Phytophthora capsici* (*P. capsici*) infection on the potential targets-silenced *N. benthamiana* leaves (n = 10). Yellow circle shows the infected area of *P. capsici* on *N. benthamiana* leaves. Leaves were photographed under UV light at 48 h post-inoculation (hpi). Scale bars: 1 cm. (**B**) Analysis of lesion size data represent mean ± standard error (SE). ns, no significant difference. (**C**) Analysis of the relative biomass on *N. benthamiana* leaves caused by *P. capsici*. Relative biomass was determined by genomic DNA (gDNA)-based quantitative PCR (qPCR). The experiment was repeated three times with similar results. The asterisk indicates that there is a statistically significant difference (* *p* < 0.05, ** *p* < 0.01, *t*-test).

**Figure 5 jof-10-00318-f005:**
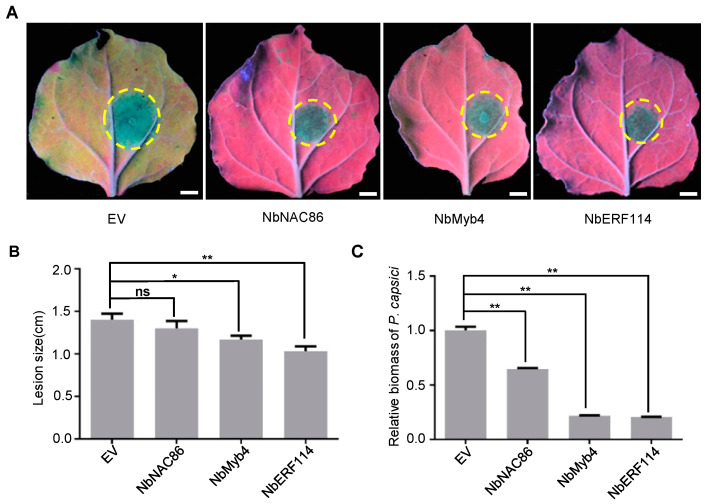
Overexpression of *NbNAC86*, *NbMyb4*, or *NbERF114* enhances resistance to *P. capsici* infection. (**A**) Disease symptoms caused by *P. capsici* infection in *N. benthamiana* leaves (n = 10), which express the EV, *NbNAC86*, *NbMyb4*, or *NbERF114* gene, respectively. Leaves were photographed under UV light. Scale bars: 1 cm. Yellow circle shows the infected area of *P. capsici* on *N. benthamiana* leaves. Lesion size (**B**) and relative biomass (**C**) of *P. capsici* in *N. benthamiana* leaves (n = 10) expressing the EV, *NbNAC86*, *NbMyb4*, or *NbERF114* gene, respectively. For relative biomass quantification, the ratio of *P. capsici* DNA compared with *N. benthamiana.* ns, no significant difference. The experiment was repeated three times with similar results. The asterisk indicates that there is a statistically significant difference (* *p* < 0.05, ** *p* < 0.01, *t*-test).

**Figure 6 jof-10-00318-f006:**
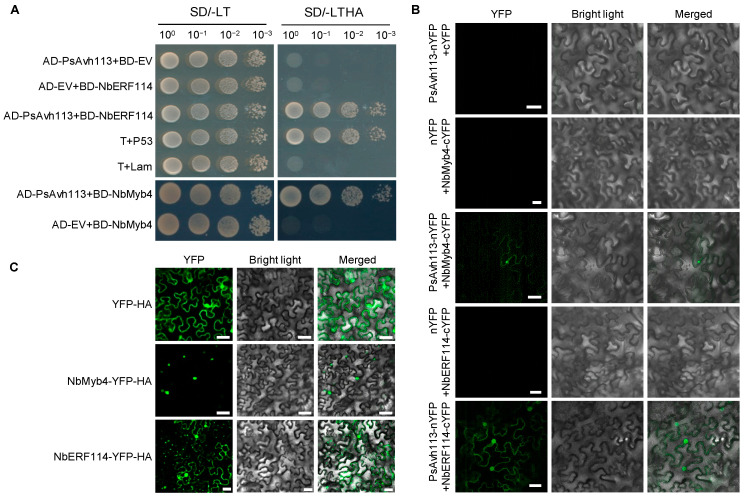
PsAvh113 interacts with NbERF14 and NbMyb4 in yeast and planta. (**A**) Y2H assay showing that PsAvh113 interacted with NbMyb4 and NbERF114 in yeast cells, respectively. The yeast strain AH109 was transformed with the bait plasmid pGBKT7 (BD) carrying NbMyb4 or NbERF114 together with the prey plasmid pGADT7 (AD) carrying PsAvh113. Transformants were selected on a minimal medium. −TL and −TLHA indicate SD/−Trp-Leu and SD/−Trp-Leu-His-Ade dropout plates, respectively. The ability to grow on −TLHA plates indicates an interaction between two proteins. T+P53 was used as the positive control, and T+Lam was used as the negative control. (**B**) Bimolecular fluorescence complementation (BiFC) assay verifying interactions of PsAvh113 with NbNAC86 and NbERF114 in leaf epidermal cells of *Nicotiana benthamiana* leaves, respectively. The experiment was repeated twice with similar results. (**C**) Subcellular localization of NbMyb4 and NbERF114 in *N. benthamiana* leaves based on A. *Agrobacterium*-mediated transient expression. Fluorescence was detected in epidermal cells of infiltrated tissues by confocal microscopy at 48 hpi. Scale bars in (**B,C**), 50 μm.

## Data Availability

The data that support the findings of this study are available in the Appendix A of this article.

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
