# Peer review of "Phytophthora sojae Effector PsAvh113 Targets Transcription Factors in Nicotiana benthamiana"

_jof, 2024, doi:10.3390/jof10050318_

Round 1

Reviewer 1 Report

In this study, the authors reported an effector PsAvh113 that showed severe viral symptoms in N. benthamiana. To elucidate its potential role in regulation immunity, they further conducted a RNA-seq analysis using the leaves with and without PsAvh113, and identified three DEG genes (NbNAC86, NbMyb4, and NbERF114) that could interact with PsAvh113 and positively manipulated the resistance to P. capsici in N. benthamiana. Collectively, the finding will advance our knowledge on identifying molecular targets of pathogen effectors. The manuscript was well-written and organized, and the data were well-interpreted. However, I still have some minor comments for authors to further improve the manuscript.

1)      Line 213, In Figure 1B, what are the “FDR” and “FC”? They should be interpreted.

2)      Line 261, “To further reveal whether PsAvh113 directly causes some EDGs,” Change “EDGs” to “DEGs”.

3)      Line 270, “NbERF114 (NbD040014) is an ERF113-like protein”. Change “ERF113” to “ERF114”.

4)      Line 281, It would be beneficial to include a graph in Figure 3 that illustrates the fold change of these three genes in the RNA-seq data. This visual representation will enhance the understanding of the data and its implications.

5)      Line 282, Change “Avh113” to “PsAvh113”.

6)      Line 339, “We performed the Y2H assay (Figure 5A).” “Figure 5A” should be “Figure 6A”, please double check it.

7)      It looks like the letter “C” is missing in Figure 6.

8)      Line 556, In Figure S1, the GFP-NbNAC86, GFP-NbMyb4, and GFP-NbERF114 in the x-axis may better be written as “TRV2-GFP” because it represents the control, not overexpressing these genes.

9)      Line 562, In Figure S2, the protein marker “KDa” should be “kDa”.

10)    Line 460, 472, 474, 482, 486, 492, 510, 521, 523, “Phytophthora” should be italicized. Please double check other words in the whole main text in italics.

Author Response

Point by point response to the reviewer’s comments (JoF-2955154)

We greatly appreciate the careful review and highly constructive comments from the editor and the reviewers. All the comments were fully considered and appropriate revisions have been modified accordingly. Two revised versions with tracked changes together with a clean copy were submitted. We believe that we have addressed comprehensively all the comments and the revised manuscript is much stronger than the original submission.

Please find below detailed point-by-point responses to the reviewers’ comments concerned by the reviewers. The comments are indicated deep blue and normal fonts.

The Review Report (Reviewer 1)

Major comments

In this study, the authors reported an effector PsAvh113 that showed severe viral symptoms in N. benthamiana. To elucidate its potential role in regulation immunity, they further conducted a RNA-seq analysis using the leaves with and without PsAvh113, and identified three DEG genes (NbNAC86, NbMyb4, and NbERF114) that could interact with PsAvh113 and positively manipulated the resistance to P. capsici in N. benthamiana. Collectively, the finding will advance our knowledge on identifying molecular targets of pathogen effectors. The manuscript was well-written and organized, and the data were well-interpreted. However, I still have some minor comments for authors to further improve the manuscript.

>>> Thank you very much for the careful review. We appreciate your encouraging and excellent suggestion.

Detail comments

1) Line 213, In Figure 1B, what are the “FDR” and “FC”? They should be interpreted.

Response: Thanks for your constructive comments! In Figure 1B, FDR represents the false discovery rate, and FC represents the fold change. We have revised it in Figure legends.

2) Line 261, “To further reveal whether PsAvh113 directly causes some EDGs,” Change “EDGs” to “DEGs”.

Response: Thanks for your kind comments! We have changed the "EDGs" in line 261 to "DEGs".

3) Line 270, “NbERF114 (NbD040014) is an ERF113-like protein”. Change “ERF113” to “ERF114”.

Response: Thanks for your constructive comments! We have changed the "ERF113" in line 270 to "ERF114".

4) Line 281, It would be beneficial to include a graph in Figure 3 that illustrates the fold change of these three genes in the RNA-seq data. This visual representation will enhance the understanding of the data and its implications.

Response: Thanks for your kind comments! We have added Figure 3A to illustrate the fold change of these three genes in the RNA-seq data and changed “3A” into “3B.”

5) Line 282, Change “Avh113” to “PsAvh113”.

Response: Thanks for your constructive comments! We have revised the "Avh113" in line 282 to "PsAvh113".

6) Line 339, “We performed the Y2H assay (Figure 5A).” “Figure 5A” should be “Figure 6A”, please double check it.

Response: Thanks for your kind comments! We have changed the "Figure 5A"in line 339 to "Figure 6A".

7) It looks like the letter “C” is missing in Figure 6.

Response: Thanks for your constructive comments! We have labeled the letter “C” in Figure 6.

8) Line 556, In Figure S1, the GFP-NbNAC86, GFP-NbMyb4, and GFP-NbERF114 in the x-axis may better be written as “TRV2-GFP” because it represents the control, not overexpressing these genes.

Response: Thanks for your kind comments! We have modified the "GFP-NbNAC86, GFP-NbMyb4, and GFP-NbERF114" on the X-axis in Figure S1 to TRV2-GFP.

9) Line 562, In Figure S2, the protein marker “KDa” should be “kDa”.

Response: Thanks for your constructive comments! We have corrected the protein marker "KDa" in Figure S2 into "kDa".

10) Line 460, 472, 474, 482, 486, 492, 510, 521, 523, “Phytophthora” should be italicized. Please double check other words in the whole main text in italics.

Response: Thanks for your kind comments! We have carefully checked and revised the "Phytophthora" in italics in Lines 460, 472, 474, 482, 486, 492, 510, 521, 523, and others.

Reviewer 2 Report

The article “Phytophthora sojae Effector PsAvh113 Targets Transcription Factors to Promote Infection in N. benthamiana” is devoted to the important and acute theme of the investigation of the interaction of P. sojae effector with N. benthamiana transcriptional factors. Authors provide and analyze a lot of information on this theme, but the article makes an ambivalent  impression since it is carelessly designed.

I have some comments:

Please, check sentences spelling “viral symptom”. In my opinion, in some cases it must be replaced with “disease symptom” or “root rot symptoms”.

Title: Please, give full latin names of species.

Abstract: This part must be clear and easy to understand, regardless of the title and the text of the manuscript. It must be clear that PsAvh113 is an effector of P. soja, NbNAC86, NbMyb4, and NbERF114 are transcriptional factors of N. benthamiana. Lines 18-19 - symptoms of what disease?

Lines 20-21 - I cannot find “viral symptoms” in this article.

Introduction

Please, write Phytophthora sp. if you don’t know species (and use italics for latin names).

It is important to write about the significance of parameters which were investigated.

Mat Met

In total: provide model, brand, country of all equipment used. Insert © then it is necessary. Insert references on all methods used.

Results

Figures 3-5: :”The asterisk indicates that there is a statistically significant difference (*P<0.1, **P<0.01, T-test).” Difference from what parameter?

Discussion: Please, provide clear and concise conclusions on your work. 

Line 73: “RxLR effectors to suppress the programmed cell death in plants” RxLR can have not only this function.

Lines 60-61: “However, plants do not just sit back and wait.” Seriously?

Author Response

The Review Report (Reviewer 2)

Major comments

The article “Phytophthora sojae Effector PsAvh113 Targets Transcription Factors to Promote Infection in N. benthamiana” is devoted to the important and acute theme of the investigation of the interaction of P. sojae effector with N. benthamiana transcriptional factors. Authors provide and analyze a lot of information on this theme, but the article makes an ambivalent impression since it is carelessly designed.

>>> Thank you very much for the constructive comments. We appreciate your encouraging and excellent suggestion. As Reviewer 3 also suggested, we have revised the title.

I have some comments:

1) Please, check sentences spelling “viral symptom”. In my opinion, in some cases it must be replaced with “disease symptom” or “root rot symptoms”.

Response: Thanks for your constructive comments! We have revised part of the text from "viral symptom" to "disease symptom".

2) Title: Please, give full latin names of species.

Response: Thanks! We have changed the article's title to "Phytophthora sojae Effector PsAvh113 Targets Transcription Factors to Promote Infection in Nicotiana benthamiana ".

3) Abstract: This part must be clear and easy to understand, regardless of the title and the text of the manuscript. It must be clear that PsAvh113 is an effector of P. soja, NbNAC86, NbMyb4, and NbERF114 are transcriptional factors of N. benthamiana. Lines 18-19 - symptoms of what disease?

Response: Thanks for your constructive comments! In lines 18-19, we think that PsAvh113 expression enhances the viral symptoms in N. benthamiana.

4) Lines 20-21 - I cannot find “viral symptoms” in this article.

Response: Thanks for your careful review! In our previous research, we showed that PsAvh113 expression increased viral symptoms. Therefore, we directly cited it in this manuscript.

Xiaoguo Zhu, et al. Phytophthora sojae effector PsAvh113 associates with the soybean transcription factor GmDPB to inhibit catalase-mediated immunity. Plant Biotechnology Journal 2023, 21, (7), 1393-1407.

Introduction

5) Please, write Phytophthora sp. if you don’t know species (and use italics for latin names).

Response: Thanks for your constructive comments! We have changed the term "Phytophthora" in the article to "Phytophthora sp.”.

It is important to write about the significance of parameters which were investigated.

Mat Met

6) In total: provide model, brand, country of all equipment used. Insert © then it is necessary. Insert references on all methods used.

Response: Thanks for your suggestions! We have added © after the model, brand, and country of the equipment mentioned in the article.

Results

7) Figures 3-5: “The asterisk indicates that there is a statistically significant difference (*P<0.1, **P<0.01, T-test).” Difference from what parameter?

Response: Thanks for your careful review! We have made changes to Figure 3-5.

8) Discussion: Please, provide clear and concise conclusions on your work.

Response: Thanks for your constructive comments! We have added this part in the discussion section.

Detail comments

9) Line 73: “RxLR effectors to suppress the programmed cell death in plants” RxLR can have not only this function.

Response: Thanks for your kind comments! We agree with your opinion. This is one of the pathogenic mechanisms.

10) Lines 60-61: “However, plants do not just sit back and wait.” Seriously?

Response: Thanks for your constructive comments! We have revised this sentence. in lines 67-68. "However, plants have evolved diverse strategies to cope with pathogen infection."

Reviewer 3 Report

The work deals with an effector of the pathogen Phytophthora sojae. Research on pathogen’s effectors plays an important role in plant pathology recent research.

Overall the manuscript describes interesting research, experiments are well done, exposition is not always very clear, perhaps because the reader is confused on the objective (N.bentamiana or soybean?, P sojae or P.capsici?). 

 From experiments of this work, I sort out mainly two conclusions:

1)      Expression of PsAvh113 in N. bentamiana upregulated plant genes  NbNAC86, NbMyb4, and NbERF114 that act as positive immune regulators against P. capsici infection in N. benthamiana.  The question is: does expression of PsAvh113 in N. bentamiana makes the plant more resistant to subsequent Phytophthora infections (behaving as an avirulence gene)?

2)      The effector PsAvh113 binds to the soybean transcription factor GmDPB and inhibit immunity to P. sojae infection; this work shows that effector PsAvh113 is able to bind also to two transcription factors NbMyb4, and NbERF114 involved in N. bentamiana resistance to P. capsici. Also remarking that work still to be done, authors compare too easily the two systems soybean/P.sojae and N. bentamiana/P.capsici but conclusions should not be hasty: pathogens but especially plants are different.

 Especially Methods are difficult to follow.

I attach the revised file with specific  suggestions and comments in the text. Suggested corrections are expressed by conditional.

Author Response

The Review Report (Reviewer 3)

Major comments

The work deals with an effector of the pathogen Phytophthora sojae. Research on pathogen’s effectors plays an important role in plant pathology recent research.

Overall the manuscript describes interesting research, experiments are well done, exposition is not always very clear, perhaps because the reader is confused on the objective (N.bentamiana or soybean?, P sojae or P.capsici?).

>>> Thank you very much for the careful review and constructive comments. We appreciate your encouraging and excellent suggestion.

Detail comments

From experiments of this work, I sort out mainly two conclusions:

1) Expression of PsAvh113 in N. bentamiana upregulated plant genes NbNAC86, NbMyb4, and NbERF114 that act as positive immune regulators against P. capsici infection in N. benthamiana. The question is: does expression of PsAvh113 in N. bentamiana makes the plant more resistant to subsequent Phytophthora infections (behaving as an avirulence gene)?

Response: Thanks, it is a very excellent question! I understand your concern. Our results showed that PsAvh113 expression in N. benthamiana makes the plant more susceptible to subsequent Phytophthora infections. To our knowledge, the increment in transcriptional level does not mean the enhancement of protein level; PsAvh113 may suppress their expression at the protein level via an unknown mechanism. This needs to be confirmed in future work.

2) The effector PsAvh113 binds to the soybean transcription factor GmDPB and inhibit immunity to P. sojae infection; this work shows that effector PsAvh113 is able to bind also to two transcription factors NbMyb4, and NbERF114 involved in N. bentamiana resistance to P. capsici. Also remarking that work still to be done, authors compare too easily the two systems soybean/P.sojae and N. bentamiana/P.capsici but conclusions should not be hasty: pathogens but especially plants are different.

Response: Thanks for your suggestion! I understand your concern. Given that P. sojae can only infect soybean and can’t infect N. bentamiana, we used the P. capsici-N. bentamiana model system to figure out the mechanism of the P. sojae effector. We must confirm the functions of these genes in the host plant (Soybean).

Especially Methods are difficult to follow.

Response: Thanks for your careful review! Our work mainly focused on the virulence effectors that increased disease symptoms upon the viral infection. Like those effectors, we think it could follow our method.

I have attached the revised file with specific suggestions and comments in the text. Suggested corrections are expressed by conditional.

Response: Thanks for your constructive comments; I appreciate them!

3) Title, line 3.

Delete “to Promote Infection”. As you say “further research is needed to understand their biological function in N.benthamiana”. Change “N. benthamiana” to “Nicotiana”

Response: Thanks! We have changed the article's title to "Phytophthora sojae Effector PsAvh113 Targets Transcription Factors in Nicotiana benthamiana."

4) Line 34, change “our country” to “China”.

Response: Thanks for your kind comments! We have changed "our country" to "China" in line 34.

5) Line 35-39, delete “Soybeans are high in nutritional value, providing 30 percent fat and 60 percent vegetable protein. Soybeans are an essential foundation and strategic material for the national economy and people's livelihood. China is the world's largest food importer, and soybeans account for nearly 80 percent of its imports. China has fought many invisible "soybean wars" around soybeans.” Redundant, you can eliminate all the sentence.

Response: Thanks for your constructive comments! We have deleted the text.

6) Line 58, “effector”

Response: Thanks! We deleted and revised it.

7) Line 58, 61,65, change “The pathogenic bacteria” to “pathongens”.

Response: Thanks for your suggestions! We have changed "The pathogenic bacteria" to "pathogens".

8) Line 64, change “does not” to “not always”.

Response: Thanks for your kind comments! We have changed " does not " to " not always ".

9) Line 65, change “Pathogenic bacteria” to “The latter may”.

Response: Thanks for your constructive comments! We have changed " Pathogenic bacteria " to " The latter may ".

10) Line 77, “specific RxLR effectors may trigger immune responses mediated by plant NLRs.” It is not only their main function.

Response: Thanks for your kind comments! We revised this sentence

11) Line 104. I don't manage to understand from previous literature(Shi et al, 2020, Zhu et al 2023) how this effector has been identified, it is taken from a RXLR list of P.sojae?; where does its sequence being deposited?

Response: Thanks for your careful review! Genome sequence was performed by Brett Tyler in 2006 Science, and the PsAvh113 effector came from one paper, which published in Plant Cell (Qunqing Wang et al. Transcriptional Programming and Functional Interactions within the Phytophthora sojae RXLR Effector Repertoire. The Plant Cell 2011, 23, (6), 2064-2086.) and JGI Project (http://www.jgi.doe.gov, Genome assembly P. sojae V3.0)

10) Line 107, delete “that”.

Response: Thanks for your kind comments! We have removed the "that" in line 107.

12) Line 124, specify which kind of samples have been analyzed by RNA-Seq, i.e- mock, EV and PsAvh113, so that later in the text is possible to understand what do you mean with SS113_EV, etc. Specify here what do you mean for EV, tha name of the plasmid, that it contain also GFP and all important details. For better understanding you may add an additional paragraph before this to explain the VIVE assay.

Response: Thanks for your constructive comments! We have added this decription in materials and method section." We have also added the method description of VIVE to the material method.

13) Line 123. I would put this paragraph after P. capsici inoculation and RNA isolation methods paragraphs.

Response: As suggested! we have adjusted the position of this paragraph in material method section.

14) Line 161, specify the firm of the kit used.

Response: Thanks! We added the company name (Vazyme, Nanjing, China) to the DNA extraction kit.

15) Line 261, change “EDGs” to “DEGs”.

Response: Thanks for your kind comments! We have changed " EDGs " to " DEGs ".

16) Line 295, change “susceptibility” to “susceptible”.

Response: Thanks! We have changed " susceptibility " to " susceptible ".

17) Line 297,301, change “silencing” to “silenced”.

Response: Thanks! We have changed " silencing " to " silenced ".

18) Line 313, change “their functions in” to “the functions of”.

Response: Thanks! We have changed " their functions in " to " the functions of ".

19) Line 329, control this captions: some parts repeated, some parts missing.

Response: Thanks! We have revised them.

20) Line 331, delete “under white light”.

Response: Thanks! We have removed "under white light" in line 331.

21) Line 334, delete “Scale bars in (A),”.

Response: Thanks! We have removed "Scale bars in (A)" from line 334.

22) Line 339, change “5A” to “6A”.

Response: Thank you very much! We have changed " 5A " to " 6A ".

23) Line 354, explain T+P53 and T+Lam.

Response: Thanks for good suggestion! We have explained "T+P53" and "T+Lam" in the figure note. T+P53 was used as the positive control, and T+Lam was used as the negative control.

24) Line 356, indicate C on the figure.

Response: Thanks! The "C" we have added in Figure 6.

25) Line 368, change “nuclear” to “nucleus”.

Response: Thanks! We have changed "nuclear" to "nucleus".

26) Line 374, very difficult to infer this conclusion from the figure.

Response: Thanks! I have included a higher resolution image below.

27) Line 283, change “[35]” to “[20]”.

Response: Thanks! We have changed "[35]" to "[20]" in line 283.

28) Line 397, “All three targets encode transcription factors that positively affect plant resistance to Phytophthora pathogens.” But in a different host, N.bentamiana.

Response: Thanks! We have revised this sentence.